# Epidemiology and Risk Factors of UTIs in Children—A Single-Center Observation

**DOI:** 10.3390/jpm13010138

**Published:** 2023-01-10

**Authors:** Maria Daniel, Hanna Szymanik-Grzelak, Janusz Sierdziński, Edyta Podsiadły, Magdalena Kowalewska-Młot, Małgorzata Pańczyk-Tomaszewska

**Affiliations:** 1Department of Pediatrics and Nephrology, Medical University of Warsaw, 02-091 Warsaw, Poland; 2Department of Medical Informatics and Telemedicine, Medical University of Warsaw, 02-091 Warsaw, Poland; 3Department of Pharmaceutical Microbiology, Centre for Preclinical Research, Faculty of Pharmacy, Medical University of Warsaw, 02-091 Warsaw, Poland

**Keywords:** UTI, pyelonephritis, cystitis, risk factors, etiology, children

## Abstract

Urinary tract infections (UTIs) are one of childhood’s most common bacterial infections. The study aimed to determine the clinical symptoms, laboratory tests, risk factors, and etiology of different UTIs in children admitted to pediatric hospitals for three years. Methods: Patients with positive urine cultures diagnosed with acute pyelonephritis (APN) or cystitis (CYS) were analyzed for clinical symptoms, laboratory tests, risk factors, and etiology, depending on their age and sex. Results: We studied 948 children with UTIs (531 girls and 417 boys), with a median age of 12 (IQR 5–48 months). A total of 789 children had clinical symptoms; the main symptom was fever (63.4% of patients). Specific symptoms of UTIs were presented only in 16.3% of patients. Children with APN had shown significantly more frequent loss of appetite, vomiting, lethargy, seizures, and less frequent dysuria and haematuria than children with CYS. We found significantly higher median WBC, CRP, and leukocyturia in children with APN than with CYS. The risk factors of UTIs were presented in 46.6% of patients, of which 35.6% were children with APN and 61.7% with CYS. The main risk factor was CAKUT, more frequently diagnosed in children with CYS than APN, mainly in children <2 years. The most commonly isolated bacteria were Escherichia coli (74%). There was a higher percentage of urine samples with *E. coli* in girls than in boys. Other bacteria found were Klebsiella species, Pseudomonas aeruginosa, Proteus mirabilis, and Enterococcus species. Conclusions: Patients with APN were younger and had higher inflammatory markers. Often, fever is the only symptom of UTI in children, and other clinical signs are usually non-specific. The most common UTI etiology is *E. coli*, regardless of the clinical presentation and risk factors.

## 1. Introduction

Urinary tract infections (UTIs) are one of the most common bacterial infections in childhood [1], affecting approximately 7.8% of children <19 years with urinary symptoms and/or fever [2]. The prevalence of UTIs in children varies by age, race/ethnicity, sex, and circumcision status [3,4]. UTIs are more common in boys (3.7%) than in girls (2%) in the first year of life, and they may be the first symptom of congenital anomalies of the kidneys and urinary tract (CAKUT) [3]. UTIs become more common in girls after the first year of life [1,2,4,5,6]. This might be explained by the shorter length of the female urethra, the regular colonization of the perineum by enteric organisms, high vaginal pH, and increased adhesiveness of bacteria to vaginal cells [4].

UTI classification depends on the site, episode, symptoms, and risk factors. UTIs may be divided into three different categories: acute pyelonephritis (APN), cystitis (CYS), and asymptomatic bacteriuria (ABU) [1,2].

*Escherichia coli* is the most common bacterial cause of UTIs, and it accounts for approximately 80% of UTIs in children; the prevalence of *E. coli* is higher among females than males [2,4,6]. In serotypes of *E. coli* frequently isolated in UTIs, bacterial adherence is enhanced by adhesins (including type 1 pili, P-fimbriae, and X-adhesins). These adhesins facilitate the adherence of the bacteria to the mucosal receptors in the uroepithelium, which causes the internalization of the bacterium into the epithelial cell and leads to a UTI [1,4,5,6].

Moreover, uropathogenic strains of *E. coli* have a defensive mechanism that consists of a glycosylated polysaccharide capsule that interferes with phagocytosis and complement-mediated destruction. An intracellular biofilm is formed on the uroepithelium and protects the uropathogenic *E. coli* from the host’s immune system [4,6,7].

Children’s other common species are *Enterococcus*, *Proteus*, *Klebsiella*, and *Pseudomonas aeruginosa* [8,9]. Viruses and fungi are uncommon causes of UTIs in children and are usually limited to the lower urinary tract [4,6].

Many factors are associated with an increased risk of UTIs in children. These include female sex, CAKUT, bladder and bowel dysfunction (BBD), a neurogenic bladder (NB), urolithiasis, diabetes mellites (DM), and immunodeficiency [1,10].

According to current global guidelines, UTIs in children may be difficult to diagnose, especially in children <2–3 years. In children, the key finding is fever. In infants and not toilet-trained children, fever may be the only symptom of UTIs, or it may be associated with poor feeding, failure to thrive, lethargy, irritability, loose stool, and vomiting. In toilet-trained children, classic urinary symptoms, such as frequency, dysuria, changes in incontinence habits, suprapubic or abdominal pain, and loin tenderness, are present [2,11,12,13,14,15].

This study aimed to determine the clinical symptoms, laboratory tests, risk factors, and etiology depending on the age and sex in different types of symptomatic UTIs (APN vs. CYS) in pediatric patients admitted to a tertiary pediatric hospital (the Children’s Hospital of the Medical University of Warsaw, Poland) for three years.

## 2. Materials and Methods

The study was approved by the ethics committee of the Medical University of Warsaw (AKBE/179/2018, date 8 October 2018). As it was a retrospective study, the requirement for informed consent was waived (patients whose parents or legal guardians consented to use the child’s data for scientific research upon admission to the hospital).


**Trial design**


The study was designed as a retrospective analysis trial.


**Settings and participants**


The study was performed in pediatric units and the outpatient clinic of the tertiary pediatric hospital.


**Eligibility criteria:**


Aged from 0 to 18 years;Positive urine culture;Leukocyturia ≥10 white blood cells per high power field (WBC/hpf).


**Exclusion criteria:**


Patients whose parents or legal guardians did not consent to the use of the child’s data for scientific research upon admission to the hospital.


**Methods**


Patients with positive urine cultures and diagnosed with symptomatic UTIs (APN or CYS defined according to the current guidelines [3]) were analyzed according to their clinical symptoms, laboratory tests, risk factors of UTIs, and etiology depending on their age (0–2 years, >2–7 years, and >7 years) and sex.

APN was defined as a diffuse pyogenic infection of the renal pelvis and parenchyma, pyuria ≥10 WBC/hpf, and positive urine culture, with fever (≥38 °C) with or without other clinical symptoms of a UTI. CYS was recognized as the inflammation of the urinary bladder (pyuria ≥10 WBC/hpf) and positive urine culture with or without clinical symptoms such as dysuria, frequency, urgency, malodorous urine, incontinence, hematuria, suprapubic pain, and non-specific signs in newborns and infants [16].

Patients were analyzed by their clinical status (fever and other accompanying symptoms, e.g., loss of appetite, vomiting, loose stools, abdominal pain, dysuria, Goldflam’s symptom, lethargy, seizures, and hematuria) and laboratory test results (WBC, neutrophils, CRP, leukocyturia, and urine culture). Goldflam’s positive symptom was described when severe, percutaneous pain was felt during shaking in the lumbar (kidney) regions; it occurs in APN and urothialysis [16].

In all children with UTIs, an ultrasound of the abdominal cavity was performed. Voiding cystography was performed in children with recurrent infections and suspected CAKUT. A recurrent UTI was defined as:≥2 infections in the upper urinary tract in one year or;A total of 1 UTI in the upper urinary tract and ≥1 infection in the lower urinary tract in one year or;≥3 UTIs in the lower urinary tract in one year [12,16].

The etiologies of UTIs in children were analyzed depending on the sex, age, type of UTI (APN vs. CYS), and risk factors, such as CAKUT, NB, constipation, bladder dysfunction, urolithiasis, DM, and immunosuppressive drugs. Urinalysis and urine cultures were collected via midstream or bladder catheterization. The presence of ≥10 WBC/hpf in the centrifuge urine sediment was considered pyuria [16]. The presence of leukocytes in the urine was analyzed in the following ranges: 10–25, 26–50, and the whole field of view. Significant bacteriuria in the case of urine collected from the middle stream was considered to be the growth in the urine of more than 100,000 (10^5^) colony-forming units per mL (CFU/mL) of uropathogenic bacteria, and for urine collected via bladder catheterization, it was considered when there were more than 10,000 (10^4^) CFU/mL [3].


**Statistical analysis**


The statistical evaluation of the obtained results was performed using the SAS 9.4 statistical package. A descriptive analysis of the results of the conducted study was performed. Then, using the Shapiro-Wilk test, the conformity of the studied empirical characteristics to the Gauss distribution was evaluated. Student’s *t*-tests were used to show the significance of differences for the variables that conformed to a normal distribution. Non-parametric Mann-Whitney U tests and the Chi2 test evaluated the features that did not have a normal distribution.

*p* < 0.05 was taken as the significance level in the analyses performed.

## 3. Results

Between 2015 and 2017, 10,006 urine cultures were performed at the children’s tertiary hospital, of which 1195 (11.1%) were positive. Among children with positive urine culture were diagnosed: 948 UTIs (79.3%), 202 ABU (16.9%), and 45 vulvitis, and foreskin inflammation or contamination (3.8%), as shown in Figure 1.

Then, we analyzed the children with symptomatic UTIs (531 girls and 417 boys, ratio 1.27:1). The median age of children with UTIs was 12 (IQR 5–48 months). The girls were older than the boys (median age 17, IQR 7–65 months vs. 7, IQR 3–28 months, *p* < 0.0001). In total, 548 patients (331 girls and 217 boys, median age 10, IQR 4–25 months) were diagnosed with APN, and 400 patients (200 girls and 200 boys, median age 23.5, IQR 5–84.5 months) were diagnosed with CYS. The frequency of the female sex in children with APN was significantly higher (*p* = 0.0014). The children with APN were significantly younger than the children with CYS (*p* < 0.0001). The basic data of these groups are presented in Table 1.

The largest group of patients with UTIs was observed among children 0–2 years (64.4%). UTIs were significantly more frequently diagnosed in boys than girls in the first year of life (*p* < 0.0001). In the older age groups, girls were dominated in APN and CYS.

### 3.1. Clinical and Laboratory Manifestations

Among 948 children with UTIs, 789 children had clinical symptoms. Fever was the main symptom in the analyzed group (63.4% in all children; 100% patients with APN, 13.2% with CYS). In 324 children with a UTI, fever was the only single symptom, and in 277 children, fever with other symptoms was observed. Specific symptoms of UTIs, including dysuria, a positive Goldflam’s symptom, and hematuria (11.3%, 3.5%, and 1.6%, respectively), were presented in only 155 (16.4%) patients.

In total, 325 children had non-specific symptoms of UTIs, such as loss of appetite, abdominal pain, vomiting, loose stools, and lethargy (17.9%, 14.2%, 11.3%, 11.3%, and 6.7%, respectively). Febrile seizures were observed in 10 children with UTIs.

The comparison of the clinical symptoms between children with different type of UTI showed in children with APN significantly more frequent than in CYS loss of appetite (*p* < 0.001), vomiting (*p* < 0.001), lethargy (*p* = 0.0087) and seizures (*p* = 0.0382), and less frequent dysuria (*p* = 0.0040) and hematuria (*p* = 0.013). The clinical presentations in the analyzed groups are shown in Table 1.

We found a significant difference between the children with APN and CYS in the median WBC (*p* < 0.0001), neutrophils (*p* < 0.0001), and CRP (*p* < 0.0001) variables. Leukocyturia was significantly higher in children with APN vs. CYS (*p* = 0.0018). The median values of the WBC, neutrophils, and CRP are presented in Table 1, and pyuria is shown in Figure 2.

### 3.2. Risk Factors

Risk factors for UTIs were present in 46.6% of patients, 35.6% of children with APN, and 61.7% with CYS. The main risk factor of a UTI was CAKUT (26.2%), mainly in children 0–2 years (64.1%); it was diagnosed more frequently in children with CYS than APN (*p* = 0.0023) and boys (56.4%). The most frequent anatomical abnormality was vesicoureteral reflux (VUR), diagnosed in 125 children (VUR: I–III grade in 58 children and IV–V grade in 67 children) and posterior urethral valves (PUV) in 37 boys.

Other risk factors were NB, constipation, bladder dysfunction, immunosuppressive drugs, urolithiasis, and DM. NB was diagnosed in 10 children with APN and 44 with CYS, mainly in children >7 years (53.7%), with an equal frequency in girls and boys. Constipation was observed in 24 children with APN and 29 with CYS, mainly in children >2 years (67.9%) and in girls (60.4%). Bladder dysfunction was diagnosed in 19 children with APN and 26 with CYS, mainly in children ≤7 years (80%) and girls (55.6%). Immunosuppressive treatments and DM could affect the clinical symptoms of UTIs, so in the study group, each of the 21 patients on immunosuppressive treatments and the 10 with DM were analyzed separately (APN or CYS were diagnosed based on the additional investigations and clinical presentations).

The risk factors for UTI were significantly more frequent in children with CYS than in children with APN: CAKUT (*p* = 0.0023), bladder dysfunction (*p* < 0.001), and NB (*p* < 0.001). We observed a tendency for more frequent constipation in CYS (*p* = 0.0575).

We found no significant differences in the frequency of risk factors, such as urolithiasis, DM, or immunosuppressive drugs, between children with APN and CYS.

The risk factors of UTIs in children according to their clinical presentations (APN and CYS) and age groups are presented in Table 2.

### 3.3. Etiology

The most frequently isolated bacterial strains in the urine samples were *Escherichia coli* (74.0%). Other bacteria found in our study were *Klebsiella pneumoniae* (8.7%), *Pseudomonas aeruginosa* (4.6%), *Proteus mirabilis* (4.3%), *Enterococcus cloacae* (1.9%), *Klebsiella oxytoca* (1.8%), and *Enterococcus faecalis* (1.7%); each of the other bacteria represented less than 1% of the total.

*E. coli* was the dominant bacteria in APN and CYS, being significantly more frequent in APN (84.7% vs. 59.5%, *p* < 0.0001). There was a higher percentage of urine samples with *E. coli* in girls than in boys (79.3% vs. 67.4%). The etiology of UTIs (*E. coli* and other than *E. coli* bacteria), according to sex, age, and type of UTI, are shown in Figure 3 and Figure 4.

The most frequently isolated bacterial strains were *E. coli*, *K. pneumoniae*, *Ps*. *aeruginosa*, *Enterococcus* spp, and *P. mirabilis*, both in patients without risk factors (84.5%; 6.7%; 3.0%; 1.5%, and 0.9% respectively) and in patients with risk factors (59.8%; 11.5%; 7%; 6.3%, and 8.5%, respectively), Figure 5.

## 4. Discussion

UTIs commonly occur in children, and as was estimated, 7.8% of girls and 1.7% of boys would have at least one episode by seven years of age [17,18,19].

The peak of UTIs was observed in the first year of life and another peak between two and four years of age, which corresponds to the toilet training age [19,20,21].

The incidence of UTIs during the first year of life was approximately 0.7% in girls and 2.7% in uncircumcised boys [20,21,22,23]. After one year, girls are much more likely than boys to develop a UTI [1,2,4,5,6]. Hispanic and white children have a two to fourfold higher prevalence of UTIs than black children [24,25,26,27].

Racial and ethnic differences exist in the rates of urinary tract infections in febrile infants in the emergency department. The recurrence rate of UTIs is 30 to 50% [22,24] and is especially common in girls [1,2,3,4,5,6,7,8,9,10,11,12,13,15,18,26,27].

In our group of white children hospitalized for a UTI, most of them were diagnosed with APN, the mean age was 12 months, and the girls were prevalent. As we expected, boys significantly predominated in the age group up to one year (*p* < 0.0001). In the older age groups, girls dominated among children with APN and CYS. These data were consistent with those published previously [12,14,15,20,23].

In the analyzed group, children with APN were significantly younger than the children with CYS. Most children with APN were <1 year of age without risk factors. Most children with CYS were >2 years of age with risk factors. The age distribution of these patients was associated with indications for hospitalization according to the recommendations of the Polish Society of Pediatric Nephrology [28]. Hospitalization was required more frequently among infants due to the need for monitoring and treatment and in older children with risk factors and complicated UTIs.

Unexplained fever is the most common symptom of a UTI during the first two years of life [18,19,20,29], and it may be the only presenting symptom of a UTI. The prevalence of UTIs is greater in infants with temperatures ≥39 °C than those with temperatures <39 °C. The American Academy of Pediatrics recommends that UTIs be considered in any infant or child between two months and two years of age presenting with fever without an identifiable source of infection [30].

Among 948 children, 83% presented clinical symptoms; 63% of children had a fever, and in 34% of children, fever was the only clinical sign of a UTI. Other non-specific manifestations of UTIs included irritability, poor feeding, vomiting, abdominal pain, and failure to thrive [17,18]. Our group mainly observed loss of appetite, abdominal pain, vomiting, loose stools, and lethargy. After the second year of life, the symptoms and signs of UTIs are more specific [3,11,12,18,20,29,31]. We observed dysuria, positive Goldflam symptoms, and hematuria in 11.3%, 3.5%, and 1.6% of patients, respectively.

Many authors have shown that a blood count assessment and inflammatory profile are helpful in the decision of whether to hospitalize a child with a UTI. Neutrophilia, an elevated erythrocyte sedimentation rate, elevated serum C-reactive proteins, and white blood cell casts in the urinary sediment are suggestive of APN [32]. These tests, however, have low specificity and cannot accurately differentiate CYS from APN. Children with high serum procalcitonin levels during a UTI are more likely to have APN [32,33]. Leung et al. [18], in a meta-analysis of 18 studies involving 831 children with APN and 651 children with CYS, found that a serum procalcitonin cut-off value of 1.0 ng/mL provided a good diagnostic value for discriminating APN from lower UTIs. In our study, the WBC and CRP were above the limits in all group ages and were significantly higher in children with APN compared to CYS. We did not analyze procalcitonin because of the retrospective nature of our study; procalcitonin was analyzed only for patients with severe APN and suspected urosepsis. In our group, the level of leukocyturia was significantly higher in children with APN than in children with CYS.

Currently, several urinary and blood biomarker studies are in progress [34], which would differentiate APN from CYS and could be routinely recommended in assessing and managing a child with a UTI.

*E. coli* is the most common uropathogen in UTIs worldwide, with the frequency (80 to 90%) depending on the geographical region [2,15,18,20,27]. Other frequent uropathogenic organisms are *Enterobacter aerogenes*, *K. pneumoniae*, *P. mirabilis*, *Citrobacter*, *Ps. aeruginosa*, *Enterococcus* spp., and *Serratia* spp. [11,18,22,27,34]. The frequency of microorganisms in urine cultures was reported as 6–13.3% for *Proteus* spp, 6.2% for *coagulase-negative Staphylococcus*, 4.7–5.8% for *Enterococcus* spp, 4.5–17.3% for *Klebsiella* spp, 1.5–3% for *Ps. aeruginosa*, and 6.9–26% for *Enterobacter* [23,24]. We identified *E. coli* (74.0%) as the main etiologic agent of UTIs in children regardless of age, sex, and type of UTI, followed by *K. pneumoniae*, *Ps. aeruginosa*, and *P. mirabilis*. Other bacteria (*Enterococcus cloacae*, *K. oxytoca*, and *Enterococcus faecalis*) represented less than 2% of the total. *E. coli* strains were observed more frequently in the group with APN and younger children. Ammenti et al. [10] showed that *E. coli* remains the predominant uropathogen isolated in acute community-acquired uncomplicated infections (80%), followed by *Klebsiella*, *Enterobacter*, *Proteus* spp., and *Enterococci*. Robinson et al. [11] observed that in previously well children who had not been on antibiotics, UTIs are usually due to *E. coli*, *K. pneumoniae*, *Enterobacter* spp, *Citrobacter* spp, *Serratia* spp or, in adolescent females only, Staphylococcus saprophyticus. Edlin et al. [5], in 25,418 outpatient urinary isolates, showed that *E. coli* was the most common uropathogen in UTIs, and the prevalence of *E. coli* was higher among females (83%) than males (50%, *p* < 0.001). Other common species among males were *Enterococcus* (17%), *P. mirabilis* (11%), and *Klebsiella* (10%). However, these uropathogens accounted for 5% or fewer female isolates (*p* < 0.001).

Our group found *K. pneumoniae* as an etiologic agent in 8.6% of UTIs. Similarly, Hanna-Wakim [31] showed that *E. coli* was the most common pathogen isolated among all age groups, followed by *Klebsiella* and *Proteus* spp, accounting for 79.4%, 7.9%, and 3.9%, respectively. *K. pneumoniae* infections are rare in boys except in association with anatomic or functional abnormalities in the first year of life. They are also infrequent among 2 to 13-year-old girls, but some girls experienced repeated episodes of recurrent CYS or APN [33]. Our group found *Ps. aeruginosa* in 4.6% of children with UTIs, mainly in boys and children with risk factors. Data regarding *Ps. Aeruginosa*-caused UTIs in children are scarce. Bitsori et al. [35] observed that children with Ps. aeruginosa significantly more often presented with a history of at least one previous UTI episode (*p* < 0.0001), hospitalization (*p* = 0.0001), use of antibiotics (*p* = 0.0001), malformations predisposing to UTIs (*p* = 0.004), VUR (*p* < 0.0001), an abnormal dimercaptosuccinic acid scan (*p* = 0.0003), longer hospitalization, and surgery.

We found *P. mirabilis* in 4.3% of UTIs, more commonly in boys than in girls, as other authors observed [18,26,36]. In some reports, *Proteus* spp. was reported as the main pathogen of UTIs in boys older than one year.

In children with anomalies of the urinary tract (anatomical, neurological, or functional) or a compromised immune system, *Staphylococcus aureus*, *Staphylococcus epidermidis*, *Haemophilus influenzae*, *Streptococcus pneumoniae*, *Streptococcus viridians*, and *Streptococcus agalactiae* may have been responsible for the UTIs [1,4,10,18,37,38]. In our study, *E. coli* strains were 1.4 times more frequent among children without risk factors than children with risk factors for a UTI. On the other hand, *Klebsiella* sp., *Ps. aeruginosa*, *Enterococcus* sp., *Proteus mirabilis*, and other bacteria predominated in children with risk factors. Alberici et al. [39] observed that in infants and young children of two to three years of age with a first febrile UTI, a pathogen other than *E. coli* significantly predicted high-grade VUR.

In the analyzed group, 35.6% of children with APN and 61.7% with CYS had risk factors for a UTI. This may have been due to the nature of our group, in which young children with uncomplicated APN were hospitalized according to the guidelines [28]. In contrast, older children with CYS were hospitalized when the UTI was complicated. Regardless of age, the most common risk factor for a UTI in our group was CAKUT which we found in more than ¼ of the group, more often among children with CYS (almost 1/3) than with APN (more than 1/5). CAKUT was diagnosed more frequently among boys. Other risk factors for a UTI included NB, constipation, and bladder dysfunction, which were more common in children >2 years of age. CAKUT predisposes children to UTIs because of the inadequate clearance of uropathogens. Infections associated with urinary tract malformations generally appear in children younger than five years of age. VUR, the most common urologic anomaly in children, allows bacteria to ascend from the bladder to the kidney and leads to post-void residual urine [18,20]. It is an important risk factor for recurrent UTIs and renal scarring [6,40]. Edwards [41], in a meta-analysis of over 250 articles, revealed that the prevalence of VUR was 31.1% in children who were evaluated for a UTI and 17.2% in those with normal kidneys who had voiding cystography for other indications, such as the diagnosis of hydronephrosis. Capozza et al. [42] showed a male-to-female prevalence of VUR of 3:1 in infants until six months of age and a shift that occurred at 21 to 24 months that showed an equal prevalence of VUR for both genders. This is in contrast to the marked female (92%) to male (8%) ratio, as seen in the randomized intervention for children with vesicoureteral reflux (RIVUR) trial [43]. Uncircumcised febrile male infants have a four to eightfold higher risk of a UTI than circumcised ones [44]. Despite the higher risk, most uncircumcised males do not develop a UTI [45]. In a meta-analysis, Singh-Grewal [46] showed that circumcision was associated with a significantly reduced risk of UTIs. In our group of the youngest boys (0–2 years), the UTI could have been related to phimosis, as circumcision in boys is not traditionally accepted in Poland.

In our group, NB was observed in 5.7% of children with UTIs without sex dominance. As in the case of NB, the inability to empty the bladder frequently results in urinary retention, urinary stasis, and the suboptimal clearance of bacteria from the urinary tract. Chronically elevated bladder pressure secondary to poor emptying also may cause secondary VUR, in which the elevated pressure increases the potential renal damage of pyelonephritis. Clean intermittent catheterizations help empty the NB, but catheterization may introduce bacteria to this normally sterile space. Previous studies suggest that 6–46% of children with UTIs suffer from lower urinary tract dysfunction (LUTD). Bulum [47] showed a 59% prevalence of LUTD in children with UTIs. Bowel and bladder dysfunction (BBD) is frequently seen in children with UTIs [48,49,50]. We observed constipation in 5.6% of patients and bladder dysfunction in 4.8% of patients, more frequent in girls than boys with UTIs. Urge syndrome and dysfunctional voiding are associated with post-void residual urine, which predisposes patients to UTIs [48,51]. Voiding postponement and infrequent voiding are other risk factors [52]. VUR and BBD are risk factors for recurrent UTIs, especially when they appear in combination [10,52]. These children had the highest risk of recurrent febrile or symptomatic UTIs. Other risk factors, such as immunosuppressive therapy, urinary tract stones, DM, etc., were rarely (<1%) found among our hospitalized patients with UTIs.

Our study has potential limitations because of its retrospective nature, some data was only available for some study patients, and some laboratory tests were not performed, thus limiting the power of the analysis for those variables. Other limitations were the absence of a control group and limited external validity for the isolated pathogens (which may vary from territory to territory).

## 5. Conclusions

Patients with APN are younger and have higher inflammatory markers. Fever is often the only symptom of UTIs in children; other clinical signs are usually non-specific. CAKUT and LUTD seem to be significant risks factor for UTIs in children. The most common etiology of UTIs is Escherichia coli, regardless of age, sex, and clinical presentation. Escherichia coli is also the most common etiologic agent of UTIs in children with and without risk factors, while *Klebsiella pneumoniae*, *Pseudomonas aeruginosa*, and *Proteus mirabilis* seem to be more frequent in children with risk factors.

## Figures and Tables

**Figure 1 jpm-13-00138-f001:**
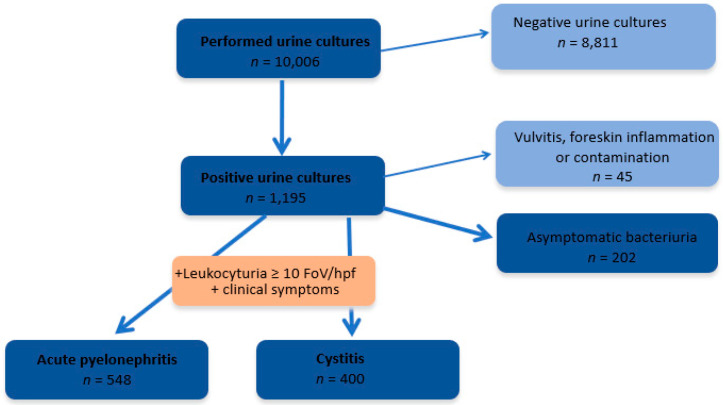
Distribution of urine culture results.

**Figure 2 jpm-13-00138-f002:**
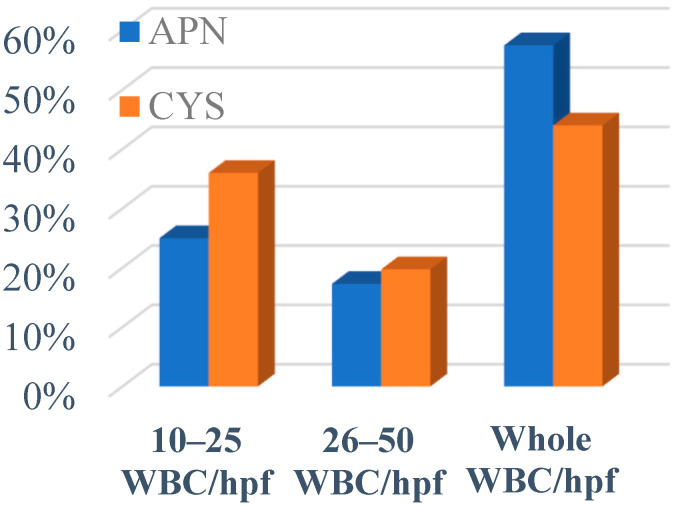
Pyuria in patients with APN and CYS. APN; acute pyelonephritis, CYS; cystitis, WBC/hpf; white blood cells per high power field.

**Figure 3 jpm-13-00138-f003:**
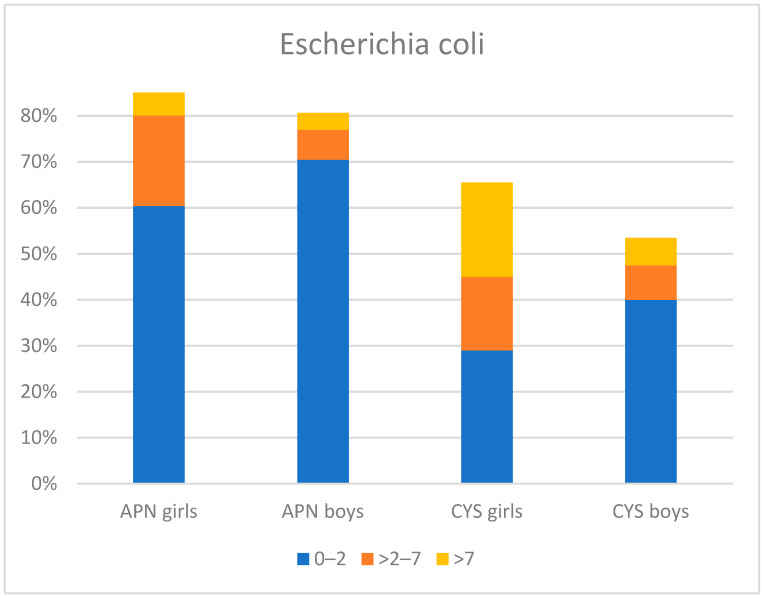
Distribution of *E. coli* strains according to sex, age, and type of UTI. APN; acute pyelonephritis, CYS; cystitis.

**Figure 4 jpm-13-00138-f004:**
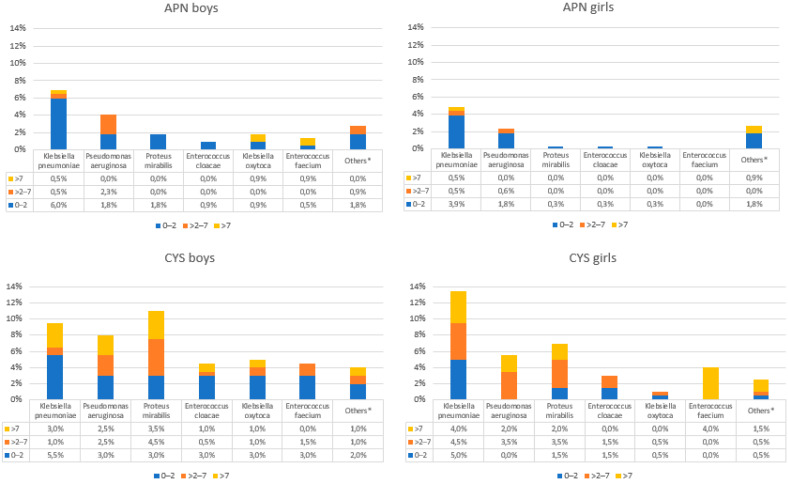
Distribution of bacterial strains according to sex, age, and type of UTI. APN; acute pyelonephritis, CYS; cystitis. * *Acinetobacter baumanii*, *Acinetobacter Iwofii*, *Citrobacter freundii*, *Enterobacter kobei*, *Klebsiella variicola*, *Morganella morgani*, *Providencia stuartii*, *Staphylococcus aureus*, *Staphylococcus saphrophyticus*, *Serratia fonticola*, *Serratia marcensces*, *Raoultella ornithinolytica*, *Raoultella planticola*.

**Figure 5 jpm-13-00138-f005:**
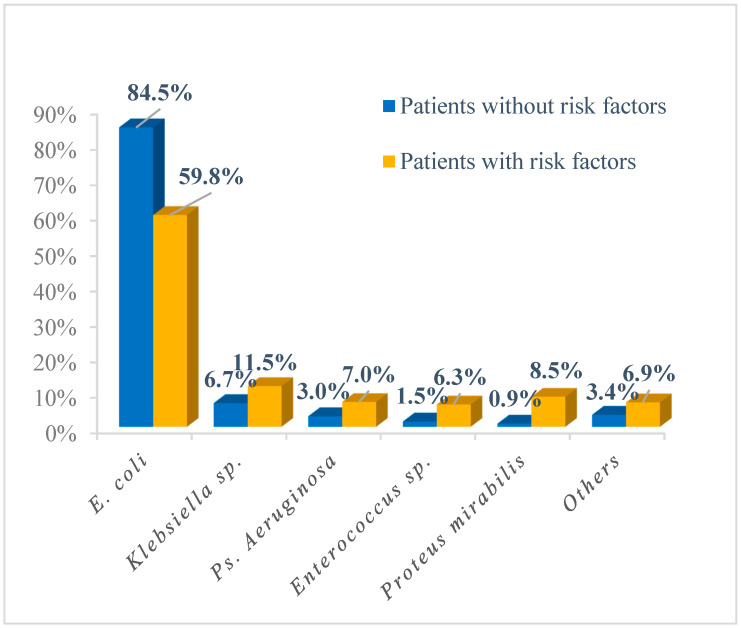
Etiology of UTIs in children with and without risk factors.

**Table 1 jpm-13-00138-t001:** Demographic data, clinical presentations, and laboratory data in children with UTIs.

	Overall *n* = 948	APN *n* = 548	CYS *n* = 400	APN vs. CYS *p*
Girls, Boys, *n*	531; 417	331; 217	200; 200	0.00014 *
Age at the onset of the UTI (months) (median (IQR)	12.0 (5.0–48.0) girls: 17 (7–65) boys: 7 (3–28)	10 (4–25)	23.5 (5–84.5)	<0.0001 *
Age category (*n*, %)				
0–2 years, *n* (%) girls, *n* (%)	611 (64.4%) 304 (49.8%)	410 (43.2%) 228 (55.6%)	201 (21.2%) 76 (37.8%)	<0.0001 *
0–1 year girls, *n* (%)	486 212 (43.6%)	318 157 (49.4%)	168 55 (32.7%)	
>2–7 years, *n* (%) girls, *n* (%)	196 (20.7%) 135 (68.9%)	97 75 (77.3%)	99 60 (60.6%)	
>7 years, *n* (%) girls, *n* (%)	141 (14.9%) 92 (65.3%)	41 28 (68.3%)	100 64 (64.0%)	
Clinical presentation (*n*, %)				
Fever ≥ 38°	601 (63.4%)	548 (100%)	53 (13.2%)	<0.0001 *
Loss of appetite	170 (17.9%)	124 (22.6%)	46 (11.5%)	<0.0001 *
Abdominal pain	135 (14.2%)	80 (14.6%)	55 (13.8%)	0.7120
Dysuria	107 (11.3%)	48 (8.76%)	59 (14.8%)	0.0040 *
Loose stools	107 (11.3%)	69 (12.6%)	38 (9.5%)	0.1374
Vomiting	107 (11.2%)	91 (16.6%)	16 (4%)	<0.001 *
Lethargy	64 (6.7%)	47 (8.6%)	17 (4.4%)	0.0087 *
Positive Goldflam’s symptom	33 (3.5%)	33 (6%)	-	**
Haematuria	15 (1.6%)	4 (0.7%)	11 (2.8%)	0.0138 *
Seizures	10 (1.1%)	9 (1.6%)	1 (0.25%)	0.0382 *
Laboratory data (median, IQR)				
WBC (×10^3^/μL)	17.6 (14.3–19.8)	17.3 (12.5–22.1)	11.21 (8.4–14.7)	<0.0001 *
Neutrophils (%)	51.2 (35.0–65.3)	55.85 (43.8–67.8)	40.6 (26.8–57.0)	<0.0001 *
CRP (mg/dL)	3.85 (0.9–8.3)	6.95 (3.8–15.3)	0.85 (0.5–2.1)	<0.0001 *

* *p* < 0.05. ** Goldflam’s symptom is characteristic of APN, and not present in CYS; therefore, these data were not compared. UTI; urinary tract infection, APN; acute pyelonephritis, CYS; cystitis, ABU; asymptomatic bacteriuria, CRP; C-reactive protein, WBC; white blood cells, WBC/hpf; white blood cells per high power fields.

**Table 2 jpm-13-00138-t002:** Risk factors of UTIs in children, according to their clinical presentations (APN and CYS) and age groups.

Children with UTIs										
Patients	Overall	APN	CYS	*p*	0–2 years		>2–7 years		>7 years	
*n*	948	548	400		611		196		141	
					APN 410	CYS 201	APN 97	CYS 99	APN 41	CYS 100
Risk factors, *n* (%) girls, *n*	442 (46.6%) 225	195 (35.6%) 117	247 (61.7%) 108	<0.0001 *	202		138		102	
CAKUT, *n* (%) girls, *n*	248 (26.2%) 108	123 (22.4%) 67	125 (31.2%) 41	0.0023 *	159 (26.0%)		58 (29.6%)		31 (22.0%)	
Neurogenic bladder, *n* (%) girls, *n*	54 (5.7%) 24	10 (1.8%) 6	44 (11.0%) 26	<0.0001 *	5 (0.8%)		20 (10.2%)		29 (20.6%)	
Constipation, *n* (%) girls, *n*	53 (5.6%) 33	34 (6.2%) 16	29 (7.2%) 17	0.0575	17 (2.8%)		21 (10.7%)		15 (10.6%)	
Bladder dysfunction, *n* (%) girls, *n*	45 (4.7%) 25	19 (3.5%) 14	26 (6.5%) 11	0.0301 *	15 (2.4%)		21 (10.7%)		9 (6.4%)	
Immunosuppressive drugs, *n* (%) girls, *n*	21 (2.2%) 17	9 (1.6%) 8	12 (3.0%) 9	0.3698	1 (0.2%)		12 (6.1%)		8 (5.7%)	
Urolithiasis, *n* (%) girls, *n*	11 (1.2%) 7	5 (0.9%) 4	6 (1.5%) 3	0.3610	4 (0.6%)		5 (2.5%)		2 (1.4%)	
Diabetes mellitus, *n* (%) girls, *n*	10 (1.0%) 7	5 (0.9%) 3	5 (1.2%) 4	0.6153	1 (0.2%)		1 (0.5%)		8 (5.7%)	

* *p* < 0.05; UTI; urinary tract infection, APN; acute pyelonephritis, CYS; cystitis, CAKUT; congenital anomalies of the kidneys and urinary tract.

## Data Availability

The data presented in this study are available on request from the corresponding author.

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
