# Peer review of "Epidemiology and Risk Factors of UTIs in Children—A Single-Center Observation"

_jpm, 2023, doi:10.3390/jpm13010138_

Round 1

Reviewer 1 Report

This study aims to analyse the epidemiology and risk factors for UTI in children. The authors performed therefore a retrospective single center analysis. While the topic holds interest for readers, some aspects have to be improved prior publication:

Presentation of data: in figure 1 are units missing, please add the appropriate labeling to all rows (age in years or months?). In case it is years, then the numbers are odd.

Discussion: The authors remain mostly descriptive. What are the clinical consequences of this paper? Should children/parents be educated? Where are knowledge gaps in the guideline?

Author Response

Dear Editor,

First, we thank you very much for your email of 3 November 2022. We appreciate the Reviewers' careful reading of our manuscript entitled Epidemiology and risk factors of UTI in children – single-center observation”.

Enclosed, please find our responses to the Editors’ comments.

We appreciate your assistance with resubmitting our manuscript and would be happy to answer any remaining questions.

Sincerely yours,

Maria Daniel

Hanna Szymanik-Grzelak

Janusz Sierdziński

Magdalena Kowalewska-MÅ‚ot

Edyta Podsiadły

Małgorzata Pańczyk-Tomaszewska

COMMENT: Presentation of data: in figure 1, are units missing; please add the appropriate labeling to all rows (age in years or months?). In case it is years, then the numbers are odd.

RESPONSE: We thank the Reviewer for this comment. We added “months” in the proper place.

COMMENT: Discussion: The authors remain mostly descriptive. What are the clinical consequences of this paper? Should children/parents be educated? Where are knowledge gaps in the guideline?

RESPONSE: We thank the Reviewer for this comment. Perhaps the work is not an obvious addition to the lack of knowledge or of the "discovery" kind. Still, it is a delineation of groups of patients at risk of UTI in the Polish environment and is the beginning of an analysis of the types of bacteria and their sensitivity in the Polish territory (another work related to treatment, the sensitivity of bacterial strains, is ongoing).

Reviewer 2 Report

The authors describe the clinical symptoms, laboratory tests, risk factors, and etiology of different types of UTIs in children admitted to pediatric hospitals for 3 years (n=948).

Topic of moderate interest. Good scientific accuracy and general organization of the paper. Sufficient use of English.

Some major issues are present together with several minor issues.

Line 39: ref

Line 41: ref. Besides, I suggest to specify the reason for which “UTIs become more common in girls after the first year of life”.

The authors stated “UTIs can be divided into three different categories: acute pyelonephritis (APN), cystitis 43 (CYS), and asymptomatic bacteriuria (ABU)”. Urosepsis, catheter-associated UTI, and (less tightly) also prostatitis and urethritis  are other types of UTIs.

Line 49: ref

I suggest specifying in the Introduction the reason for the association of UTIs with E.coli

Line 59: useless

Line 63-64: I suggest to move it in the Methods section. Besides, the authors should add the date and ID code of Ethics committee approval. Finally, informed consent for data publication is always required. The author should clarify all these points otherwise the paper, regardless of content, cannot be published.

Dividing the Methods section into subparagraphs is a good idea, but I would change the style of the headings to make them more easily identifiable

Line 71: I suggest to add the name and country of hospital

Why leukocyturia < 10 was used as exclusion criterion? The authors should clarify this point

I suggest to add among the limitations also the absence of control group and the limited external validity concerning the isolated germs (they can vary from one territory to another)

I suggest softening the conclusions of the study due to the significant limitations present (conditional tenses, hypothesis, “seem”)

Line 87: ref

Line 90: ref

I suggest to remark that the main diagnostic difference between APN and CYS was the presence of fever in APN. I suggest briefly discussing the possibility of APN without fever

Immunosuppressed children were included. Immunosuppression may also have affected symptoms (less pronounced or absent). The author should discuss this point.

Line 96: ref

How recurrent infections was defined? The authors should specify this point

Figure 1 and 4: the authors should improve the quality of these figures

Tables and Figures: the meaning of all abbreviations should be specified in the legend of each table and figure

Author Response

Dear Editor,

First, we thank you very much for your email of 26 December 2022. We appreciate the Reviewers' careful reading of our manuscript entitled Epidemiology and risk factors of UTI in children – single-center observation”.

Enclosed, please find our responses to the Editors’ comments.

We appreciate your assistance with resubmitting our manuscript and would be happy to answer any remaining questions.

Sincerely yours,

Maria Daniel

Hanna Szymanik-Grzelak

Janusz Sierdziński

Magdalena Kowalewska-MÅ‚ot

Edyta Podsiadły

Małgorzata Pańczyk-Tomaszewska

The authors describe the clinical symptoms, laboratory tests, risk factors, and etiology of different types of UTIs in children admitted to pediatric hospitals for three years (n=948).

Topic of moderate interest. Good scientific accuracy and general organization of the paper. Sufficient use of English.

Some major issues are present together with several minor issues.

RESPONSE: We thank the Editor for the revision. We have corrected point by point the suggestions sent by the Reviewer.

COMMENT: Line 39: ref

RESPONSE: We thank the Reviewer for this comment. We added references.  

COMMENT: Line 41: ref. Besides, I suggest to specify the reason for which “UTIs become more common in girls after the first year of life”.

RESPONSE: We thank the Reviewer for this comment and apologies for not being clear. We modified and clarified mentioned sentence in the revised manuscript (lines 42-44). We also added references.

COMMENT: The authors stated “UTIs can be divided into three different categories: acute pyelonephritis (APN), cystitis 43 (CYS), and asymptomatic bacteriuria (ABU)”. Urosepsis, catheter-associated UTI, and (less tightly) also prostatitis and urethritis  are other types of UTIs.

RESPONSE: We thank the Reviewer for this comment. In our study group, none of the children were diagnosed with prostatitis or urethritis, and based on EAU/ESPU guidelines, UTIs were divided into the above three types.

COMMENT: Line 49: ref

RESPONSE: We thank the Reviewer for this comment. We added references (line 61).

COMMENT: I suggest specifying in the Introduction the reason for the association of UTIs with E.coli.

RESPONSE: We thank the Reviewer for this comment and apologies for not being precise. We corrected mentioned sentences, and we have added a suggested explanation for the association between E. coli and UTI in the revised manuscript (lines 50-54). We added references (line 54).

COMMENT: Line 59: useless

RESPONSE:  We thank the Reviewer for this comment. We removed mentioned sentence.

COMMENT: Line 63-64: I suggest to move it in the Methods section. Besides, the authors should add the date and ID code of Ethics committee approval. Finally, informed consent for data publication is always required. The author should clarify all these points otherwise the paper, regardless of content, cannot be published.

RESPONSE:  We thank the Reviewer for this comment and apologies for not being clear. We modified and clarified mentioned sentences in the revised manuscript. We also moved them into the Methods section (lines77-80).

COMMENT: Dividing the Methods section into subparagraphs is a good idea, but I would change the style of the headings to make them more easily identifiable

RESPONSE: We thank the Reviewer for this comment. We changed the style of the headings.

COMMENT: Line 71: I suggest to add the name and country of hospital

RESPONSE: We thank the Reviewer for this comment. We added them to the manuscript (lines 73-74).

COMMENT: Why leukocyturia < 10 was used as exclusion criterion? The authors should clarify this point

RESPONSE: We thank the Reviewer for this comment. We removed this sentence  from exclusion criteria and added to eligibility criteria: “the presence of ≥10 WBC/hpf in centrifuge urine sediment”.

We explain the definition of UTI ( lines 103-108)  and “pyuria” in lines 123-124  according to polish guidelines.

UTI was diagnosed based on WBC>10/hpf  and a positive urine culture. It is known that UTI can occur without pyuria (e.g in polyuria) and also without a positive urine culture (e.g. in a renal abscess), but we found it difficult to identify such patients in our retrospective analysis.

COMMENT: I suggest to add among the limitations also the absence of control group and the limited external validity concerning the isolated germs (they can vary from one territory to another)

RESPONSE:  We thank the Reviewer for this comment; we added your suggestions (lines 388-389)

COMMENT: I suggest softening the conclusions of the study due to the significant limitations present (conditional tenses, hypothesis, “seem”)

RESPONSE:   We thank the Reviewer for this comment. We modified the conclusions in accordance with the reviewer suggestion (lines392-396).

COMMENT: Line 87: ref

RESPONSE: We thank the Reviewer for this comment. We added refference (line 108).

COMMENT: Line 90: ref

RESPONSE: We thank the Reviewer for this comment. We added reference (line108). 

COMMENT: The main diagnostic difference between APN and CYS was the presence of fever in APN. I suggest briefly discussing the possibility of APN without fever.

RESPONSE: We thank the Reviewer for this comment. According to current recommendations from several scientific societies: “Infants and children presenting with bacteriuria and fever of 38°C or higher should be considered for acute pyelonephritis/upper urinary tract infection. Infants and children who present with a fever below 38°C with lumbar pain/tenderness and bacteriuria should also be considered for acute pyelonephritis/upper urinary tract infection. All other infants and children with bacteriuria but no systemic symptoms should be considered cystitis/lower urinary tract infection” (NICE).

COMMENT: Immunosuppressed children were included. Immunosuppression may also have affected symptoms (less pronounced or absent). The author should discuss this point.

RESPONSE:  We thank the Reviewer for this comment and apologies for not being precise.

Immunosuppressive treatment and DM can affect the clinical symptoms of UTIs, so in the study group, each of the 21 patients on immunosuppressive treatment and the 10 with DM were analyzed separately (APN or CYS were diagnosed based on additional investigations and clinical presentation).

COMMENT: Line 96: ref

RESPONSE:   We thank the Reviewer for this comment. We added reference (line113).

COMMENT: How recurrent infections was defined? The authors should specify this point

RESPONSE: We thank the Reviewer for this comment and apologies for not being clear. We added the definition of recurrent UTI (lines 116-119).

COMMENT: Figure 1 and 4: the authors should improve the quality of these figures

RESPONSE: We improved the quality of the figures and sent them also in editable files.

COMMENT: Tables and Figures: the meaning of all abbreviations should be specified in the legend of each table and figure

RESPONSE:   We thank the Reviewer for this comment. We added the abbreviations.

Round 2

Reviewer 2 Report

The authors changed the paper according to my suggestions